# The Effect of *Rickettsia bellii* on *Anaplasma marginale* Infection in *Dermacentor andersoni* Cell Culture

**DOI:** 10.3390/microorganisms11051096

**Published:** 2023-04-22

**Authors:** Joseph A. Aspinwall, Shelby M. Jarvis, Susan M. Noh, Kelly A. Brayton

**Affiliations:** 1Department of Veterinary Microbiology & Pathology, Washington State University, Pullman, WA 99164, USA; joseph.aspinwall@wsu.edu (J.A.A.); shelby.jarvis@wsu.edu (S.M.J.); smnoh@wsu.edu (S.M.N.); 2Animal Disease Research Unit, Agricultural Research Service, United States Department of Agriculture, Pullman, WA 99164, USA

**Keywords:** *Rickettsia bellii*, *Anaplasma marginale*, *Dermacentor andersoni*, coinfection, superinfection, tick-borne disease

## Abstract

*Anaplasma marginale* is a tick-borne pathogen that causes bovine anaplasmosis, which affects cattle around the world. Despite its broad prevalence and severe economic impacts, limited treatments exist for this disease. Our lab previously reported that a high proportion of *Rickettsia bellii*, a tick endosymbiont, in the microbiome of a population of *Dermacentor andersoni* ticks negatively impacts the ticks’ ability to acquire *A. marginale*. To better understand this correlation, we used mixed infection of *A. marginale* and *R. bellii* in *D. andersoni* cell culture. We assessed the impacts of different amounts of *R. bellii* in coinfections, as well as established *R. bellii* infection, on the ability of *A. marginale* to establish an infection and grow in *D. andersoni* cells. From these experiments, we conclude that *A. marginale* is less able to establish an infection in the presence of *R. bellii* and that an established *R. bellii* infection inhibits *A. marginale* replication. This interaction highlights the importance of the microbiome in preventing tick vector competence and may lead to the development of a biological or mechanistic control for *A. marginale* transmission by the tick.

## 1. Introduction

*Anaplasma marginale* is an obligate intracellular bacterium that parasitizes red blood cells and is the causative agent of bovine anaplasmosis, which is endemic in temperate, tropical, and subtropical areas around the globe [1]. Infection leads to anemia, jaundice, fever, weight loss, and, in some instances, death [2]. After the initial acute infection, *A. marginale* typically establishes lifelong persistence, making the bovine host a reservoir for ongoing transmission [3]. Feeding ticks acquire *A. marginale* in the bloodmeal and then the bacteria infect the tick midgut before spreading to the salivary glands, from where it can be transmitted to a new bovine host during a subsequent bloodmeal. This bacterium has been estimated in previous work to cost more than USD300 million annually to the US cattle industry, and over USD800 million across Latin America [1,4]. Despite the prevalence of bovine anaplasmosis, few effective treatment options exist for *A. marginale*. The current strategies for bovine anaplasmosis treatment and prevention include antibiotics, vaccines, and acaricides, which are not only ineffective, but in the case of the latter also have severe impacts on the environment [5,6,7]. Because control strategies for bovine anaplasmosis suffer from inefficiencies, we began to explore the relationship of *A. marginale* with the tick microbiome, with the long-term goal of being able to interfere with pathogen transmission.

In our previous study, we demonstrated that after antibiotic treatment of a cohort of F0 generation *D. andersoni* ticks, the F1 generation had a higher proportion of *Rickettsia bellii* in their microbiome than the control group of F1 ticks reared from an F0 cohort from the same population that were not exposed to antibiotics [8]. When F1-generation ticks were fed on an *A. marginale*-infected animal the cohort resulting from the treated F0 ticks acquired fewer *A. marginale* than the control F1 ticks. Both the treated group and the control acquired fewer *A. marginale* than a *D. andersoni* colony collected from a different location, which was negative for *R. bellii*. *Rickettsia bellii* is an obligate intracellular bacteria known to colonize more than 25 species of ticks [9]. Although many species of *Rickettsia* can cause vertebrate disease, *R. bellii* has never been associated with disease. Interestingly, tick endosymbionts within the genus have been shown to impact the maintenance of pathogens within tick hosts [10]. *Wolbachia pipientis*, also a Rickettsiales, has been shown to impact the maintenance of a wide range of pathogens within their insect hosts [11,12]. Additionally, *Rickettsia buchneri*, an endosymbiont of *Ixodes scapularis*, has recently been shown to reduce the infection of tick cells by multiple rickettsial pathogens, including *Anaplasma phagocytophilum* [13]. Our previous work indicates a correlation between the presence of *R. bellii* in the tick microbiome and reduced acquisition of *A. marginale*; however, the specifics of the interaction are unclear. It is unknown whether *R. bellii* directly causes the decrease in *A. marginale*, if its manipulation of the host causes a less hospitable environment for *A. marginale*, or if the presence of *R. bellii* decreases the prevalence of other organisms in the microbiome that facilitate *A. marginale* infection. This study aims to isolate the interaction among *A. marginale*, *R. bellii*, and the *D. andersoni* host cell from components of the microbiome that might be affecting the interaction and in doing so, reduce the number and type of complex interactions that might lead to the observed correlation in the previous work. We do so by conducting a series of mixed infections in tick cell cultures (DAE100 cells) with both *R. bellii* and *A. marginale* and examining different time points in infection, as well as different infection conditions, in order to narrow down a potential mechanism of action for further examination.

## 2. Materials and Methods

### 2.1. DAE100 Cell Maintenance

An embryonic cell line, DAE100, from the Rocky Mountain wood tick, *D. andersoni,* was used in these experiments. DAE100 cells were maintained in the presence of 15 mL of L15B complete media under similar conditions to those described previously in the absence of gentamicin [14,15]. Less than 24 h before infection, DAE100 cells were resuspended in fresh media. Cells were then quantified using hemocytometry and diluted to the appropriate concentration for the following experiments.

### 2.2. Bacteria Preparation

*Anaplasma marginale* strain St. Maries was grown in DAE100 cells cultured in L15B media buffered as described by Solyman et al. at 34 °C [15,16]. The media was changed two times per week. The infected cultures were maintained in Greiner Bio-One CELLSTAR™ TC Treated T-75 cell culture flasks (Sigma Aldrich, St. Louis, MO, USA) for approximately two weeks or until 90% of DAE100 cells were infected with *A. marginale*. The cells were then resuspended in media, transferred to 50 mL conical tubes, and centrifuged at 2500 RCF to pellet the cells. The supernatant was removed from the tubes and the pellet resuspended in sucrose–phosphate–glutamate buffer (SPG) [17]. The cell culture solution was then sonicated at 30% amplitude in 15 s increments until 80% of the host cells were lysed, as visualized on wet mount slides. The bacteria in SPG were then passed through a 5 µm filter to remove host cell debris and frozen at a rate of 1 °C per minute to −80 °C. An aliquot of *A. marginale* in SPG was then thawed and the DNA was extracted and used to calculate the concentration of *A. marginale* with qPCR (Table 1).

*Rickettsia bellii* RML369C containing the pRAM18 dSGA plasmid expressing green fluorescent protein (GFP) under streptomycin and spectinomycin selection was used for all experiments [17]. *Rickettsia bellii* was grown in DAE100 cells cultured in L15B complete media under the DAE100 growth conditions described above. The flasks were harvested when approximately 90% of the host cells were no longer adherent. The cells were resuspended in L15B and transferred to microcentrifuge tubes. *Rickettsia bellii* was then processed as above except cells were pelleted by centrifugation at 25,000 RCF, frozen in SPG, and quantified in the same manner as described for *A. marginale*.

### 2.3. Coinfections

Twenty-four-well plates were seeded with 5 × 10^5^ DAE100 cells in 500 µL of L15B complete media and were incubated at 34 °C [16]. Less than 24 h later, the L15B complete media was replaced with buffered L15B, along with *A. marginale* at an MOI of 50 and *R. bellii* at an MOI of one or ten, depending on the experiment. To accomplish this, stocks of *A. marginale* and *R. bellii* suspended in SPG were thawed at 37 °C for 3 min, centrifuged at 25,000 RCF for 10 min, resuspended in buffered L15B, and diluted to two times the final concentration. Two hundred and fifty microliters of the appropriate bacterial suspension was added to each well, and additional buffered L15B was added to bring each well to a final volume of 500 µL. The plates were then spun at 200 RCF for 5 min to bring the extracellular bacteria into proximity with the DAE100 cells. The plates were incubated at 34 °C for 1 h in a sealed EZ Campy container system (Fisher Scientific, Waltham, MA, USA) to maintain a microaerophilic environment appropriate for *A. marginale* growth. At this point, the media was removed and replaced with media containing 50 µg/mL of gentamicin to synchronize host cell infection by killing the extracellular bacteria, and the plates were again incubated in the Campy container system along with sachets for 1 h. The media containing gentamicin was replaced with buffered L15B complete media and fresh Campy sachets were added to the container. One plate was removed from the Campy containment system every 24 h and used for further analysis, and new Campy sachets were added to the system.

### 2.4. Superinfection

The quantified DAE100 cells were used to seed three Greiner Bio-One CELLSTAR™ TC Treated T-25 cell culture flasks (Sigma Aldrich) with 5 mL of L15B complete media, each with more than 0.4 times the total number of DAE100 cells needed for the infection. Less than 24 h later, the media was replaced in the flasks. Two flasks received 5 mL of media containing *R. bellii* at an MOI of 1, while the third received just 5 mL of L15B complete media. All flasks were centrifuged at 200 RCF for 5 min and incubated for one hour at 34 °C. The media was then replaced with fresh L15B containing 50 µg/mL of gentamicin and again incubated for 1 h under the same conditions to synchronize host cell infection by killing the extracellular bacteria. The media was again replaced with L15B complete media containing 100 µg/mL of both spectinomycin and streptomycin to maintain the *R. bellii* plasmid, and the flasks were held for 48 h at 34 °C. At this point, the media was again replaced, the cells were resuspended from the flask with mechanical agitation, and the *R. bellii*-infected cells were combined. The cells were quantified via hemocytometer and used to seed 24-well plates with 5 × 10^5^ DAE100 cells in 500 mL of media. From this point on, strep and spec were removed from media to prevent effects on *A. marginale* growth. Less than 24 h later, the appropriate wells were infected with *A. marginale*, and all plates were centrifuged and treated with gentamicin using the strategy described above for coinfection. Samples were collected, the Campy sachets were replaced at 24 h increments, and the samples were then used for further analysis.

### 2.5. RNA Extraction

The wells were rinsed once with buffered L15B complete media before the RNA was extracted using the Quick-RNA Microprep Kit (Zymo Research, Irvine, CA, USA). This was followed by a second DNA-digestion step in solution using the TURBO DNase kit (Invitrogen, Waltham, MA, USA) and concentrated using the RNA Clean & Concentrator kit (Zymo Research). The RNA samples were used to make cDNA in 10 µL reactions using the VERSO cDNA Synthesis kit (Thermo Fisher, Waltham, MA, USA).

### 2.6. RT-qPCR

The cDNA was used in quantitative PCR reactions using primers corresponding to *D. andersoni* GAPDH, *A. marginale msp5*, and *R. bellii rpoB*, as listed in Table 1. All reactions used Sso Advanced Universal SYBR qPCR super mix (Bio-Rad, Hercules, CA, USA). The bacterial cycle threshold (Ct) for all samples was normalized to *D. andersoni* from the same sample. For comparison of *R. bellii* growth, the conditions were normalized to the host RNA and then to the average of the 24 h time points for the MOI of one growth condition. Comparisons of DAE100 RNA run on separate plates were normalized using the *D. andersoni* positive control. The same positive controls were used for all plates within an experiment. All samples were run in technical duplicate, and the mean Ct value was used for further analysis.

### 2.7. Validation and Statistics

All coinfections and superinfections were paired with independent *R. bellii* and *A. marginale* infections under otherwise-identical conditions. All time points were grown in triplicate, and all qPCR quantifications were executed in duplicate. The bacterial growth curves were normalized to the host cell DNA from the same sample and to the 24 h time point from the same growth condition. Differences in bacterial growth were assessed using two-way ANOVA analysis, and significant differences between growth conditions were analyzed post hoc using Tukey’s test or Šídák’s multiple-comparison test depending on the number of conditions being compared (see Appendix A). The twenty-four-hour time point comparisons were normalized to the host cell RNA and then to the average of the 24 h time points for the single-infection controls. These were analyzed using a two-tailed *t*-test. For the DAE100 growth curves, differences were assessed in the same manner as the bacterial growth curves. For comparison of *R. belii* growth, significance was assessed in the same manner as DAE100 growth curves.

## 3. Results

To understand the dynamics of the interaction among *R. bellii*, *D. andersoni* cells, and *A. marginale*, we conducted a series of three infections of DAE100 cells with both *A. marginale* and *R. bellii*. In the first two infections, 5 × 10^5^ DAE100 cells were concurrently infected with *A. marginale* and *R. bellii*. The only variation between these infections was the amount *R. bellii*, with an MOI of one for the first experiment (low-*R. bellii* coinfection) and ten for the second (high-*R. bellii* coinfection). In the third experiment, 5 × 10^5^ DAE100 cells were first infected with *R. bellii* at an MOI of 1, then three days later infected with *A. marginale* (superinfection) (Figure 1). All experiments used an MOI of 50 for *A. marginale*. These experiments address the effect of different amounts of *R. bellii* on *A. marginale* growth, as well as the impact of established *R. bellii* infection in the host cell as compared to concurrent infection by the two bacteria.

### 3.1. Coinfection with an R. bellii MOI of 1

In the coinfection experiment with a low MOI of *R. bellii*, *A. marginale* did not have a significant growth defect during coinfection compared to the *A. marginale* single-infection control (Figure 2A). There was, additionally, no difference in *R. bellii* replication in the coinfection compared to the *R. bellii* single-infection control (Figure 2B). Interestingly, however, when the 24 h time points for *R. bellii* and *A. marginale* in the coinfection were compared to their respective single-infection controls, *A. marginale* had a significant decrease in the coinfection (Figure 2C), with a mean difference of 21% across the three replicates. For *R. bellii*, there was not a significant difference in the amount of *R. bellii* present at the 24 h time point between the coinfection and the control (Figure 2C, Appendix A). This indicates that fewer *A. marginale* organisms were able to enter the host cells in the presence of *R. bellii*, but once internalized, their growth was the same.

### 3.2. Coinfection with an R. bellii MOI of 10

For the high-*R. bellii* coinfection, mixed infection did not affect the replication of either *A. marginale* or *R. bellii* (Figure 3A,B). Interestingly, at the higher dose, *R. bellii* decreased in abundance over the four days. This occurred in both the presence and absence of *A. marginale*. A similar trend to the MOI of one experiment was observed with a lower amount of *A. marginale* present at the 24 h time point of the coinfection compared to the control, with a mean difference of 14%; however, the difference was not statistically significant (Figure 3C).

### 3.3. Superinfection

For the superinfection of *A. marginale* into *R. bellii*-infected cells, *A. marginale* replication was significantly impacted at the 96 h time point, and a marginally significant effect (*p* = 0.07) was observed at the 72 h time point (Figure 4A, Appendix A). Similar to the high-MOI coinfection, *R. bellii* did not grow in either the control or the combined infection, and there was no significant difference between the two conditions at any of the four time points (Figure 4B). When the amount of *R. bellii* and *A. marginale* in the combined infection was compared to controls, both *R. bellii* and *A. marginale* showed a significant change, with *A. marginale* having an average of 27% less in the coinfection than the control and *R. bellii* having an average of 28% more (Figure 4C, Appendix A).

### 3.4. DAE100 Growth Curves

To control for host cell death as a contributing factor to bacterial replication, or the lack thereof, the growth curve for DAE100 cells was plotted based on the quantity of DAE100 cell RNA from the three experiments above and their controls. In the coinfection with the lower MOI of *R. bellii*, DAE100 cells had slow growth across the 4-day period (Figure 5A). In the high-*R. bellii*-MOI coinfection, both the *A. marginale* and combined infection had little to no growth; however, the *R. bellii* control showed a significant decrease at the 72 and 96 h time points when compared to the other two growth conditions at a *p* value below 0.05 (Figure 5B, Appendix A). Finally, the superinfected DAE100 cells had a similar lack of growth with the high-*R. bellii* coinfection and no significant difference between the coinfection and either control (Figure 5C).

### 3.5. R. bellii Growth Comparison

The amount of *R. bellii* used in the experiments is a major variable. To obtain a more complete view of its growth dynamics, *R. bellii* growth was compared among the three different experiments. As discussed in the previous sections, the amount of *R. bellii* decreases over the four days of the experiment in both the high-MOI coinfection and the superinfections. Even with the decrease in *R. bellii*, these two conditions still contain significantly more *R. bellii* than the MOI of one coinfection at all time points for the superinfection and at the first two time points for the MOI of ten coinfection (Figure 6, Appendix A).

## 4. Discussion

We previously reported that *D. andersoni* ticks that contained *R. bellii* in their microbiome acquired fewer *A. marginale* and that the amount of *R. bellii* negatively correlated with the amount of *A. marginale* acquired [8]. To explore this observation further in the absence of the potentially confounding variable(s) of other microbiome components, we examined *A. marginale* infection of DAE100 cells with varying *R. bellii* infection levels and timing. Stated another way, we conducted coinfections of DAE100 cells with two different amounts of *R. bellii* while holding the amount of *A. marginale* constant. In the third experiment, we introduced *A. marginale* into cultures with established *R. bellii* infections. Our results showed that in the presence of *R. bellii*, *A. marginale* had a reduced ability to establish infection in DAE100 cells (Figure 2C, Figure 3C, and Figure 4C). Interestingly, if *A. marginale* and *R. bellii* were introduced as coinfections, the *A. marginale* growth was unaffected (Figure 2A and Figure 3A), i.e., *A. marginale* exhibited the same rate of replication when co-cultured with *R. bellii* as when grown on its own as a single infection in DAE100 cells. However, if *A. marginale* was introduced to the culture after *R. bellii* was established, *A. marginale* had a reduced capacity to establish infection and replicate, leading to a decline in bacterial numbers over time (Figure 4A,C). These results corroborate our earlier tick microbiome observations and identify the effects of *R. bellii* on both the early infection and longer-term growth of *A. marginale*.

The reduced levels of *A. marginale* at the 24 h time point across all experiments suggest that when *A. marginale* is grown in coinfections, as compared to when it is grown on its own, *R. bellii* inhibits the early phases of host cell infection. This could be due to the ability of *R. bellii* to manipulate the host cytoskeleton, preventing *A. marginale* from entering the host cell, or to disruption of morula formation. It is also possible that the presence of *R. bellii* triggers a response in the host cell that negatively impacts *A. marginale* within the first 24 h of infection. In the absence of an established *R. bellii* infection, however, this effect was insufficient to prevent *A. marginale* replication once it had gained entry to the cell.

When *R. bellii* infection was already established in DAE100 cells when *A. marginale* was introduced, not only were fewer *A. marginale* able to establish in the culture, but the *A. marginale* that did establish did not replicate. The impaired growth could be a result of the same mechanisms described above; however, it could be attributed to additional causes, such as host nutrient sequestration or mechanical disruption of the morula by motile *Rickettsia* within host cells infected by both bacteria [17]. Whatever the cause, the established infection by *R. bellii* more accurately represents the natural interaction of these two bacteria. *R. bellii* is a tick endosymbiont capable of vertical transmission in the tick, while *A. marginale*, though capable of transstadial transmission, is primarily transmitted between ruminants by male ticks that both acquire and transmit *A. marginale* while searching for females [8,18,19,20]. Thus, when these bacteria interact, *R. bellii* is likely established in the tick microbiome well before the introduction of *A. marginale*.

Given the nature of infecting cells with two different bacteria, one primary concern is the potential of either bacterium to negatively impact the host cell viability and, consequently, impact the viability of the other bacteria in culture. To examine the host cell viability, growth curves for the DAE100 cells were plotted. All bacterial growth curves were also normalized to the host cell RNA from the same sample. These two strategies allowed for the measurement of bacteria relative to the host cell population, which should account for host cell death that does not correlate to the presence of the measured bacteria. The separate DAE100 cell growth curves, in turn, aided in identifying instances in which the host cells were significantly affected by bacteria. Only the high-MOI coinfection showed a significant change in the DAE100 cell growth between the experimental conditions, and only at the 72 and 96 h time points of the *R. bellii* single-infection control. The other two conditions, however, tracked closely together. This may indicate that *A. marginale* has a positive effect on host cell survival in the presence of *R. bellii*. Interestingly, this same phenotype was not seen in the *R. bellii* control in the superinfection experiment, suggesting that the DAE100 cell death seen in the high-MOI coinfection experiment may be a result of the high *R. bellii* MOI introduced in that experiment.

With the exception of the low-MOI coinfection, *R. bellii* did not replicate effectively. In both the high-*R. bellii*-MOI coinfection and the superinfection, *R. bellii* had a twofold reduction during the four-day experiment. This phenotype was consistent in the single-infection controls, as well as the mixed infections, and was not reflected in a decrease in the number of DAE100 cells. It is also worth noting that although *R. bellii* did not replicate, this did not correlate to a lack of growth for *A. marginale* in the high-*R. bellii*-MOI coinfection. *Rickettsia bellii* moderating its infection level in tick cells would explain such a phenotype. This would be advantageous to an endosymbiont that relies on the reproduction of the tick host for transmission. It is also possible that all the host cells were infected by the 24 h time point and that the *R. bellii* was no longer able to replicate effectively under those conditions.

The comparison of the relative amount of *R. bellii* between growth conditions is also a potentially illuminating factor in the bacterial interaction. In both the high-MOI coinfection and superinfection, *R. bellii* did not grow over the course of the experiment; however, in both conditions, the amount of *R. bellii* present was higher than in the MOI of one coinfection (Figure 6). Moreover, the superinfection had a higher amount of *R. bellii* than even the high-MOI coinfection. Though the presence of *R. bellii* during *A. marginale* introduction is identified as a factor associated with the establishment of infection, it is possible that the higher level of *R. bellii* is a major contributing factor to the lack of *A. marginale* growth seen under this condition. In addition, the amount *R. bellii* at the 24 h time point in the superinfection was higher than would be expected for the 96 h time point for *R. bellii* (the amount of time that had passed since *R. bellii* infection at an MOI of one). This may be an artifact of the difference in strategies in the separate experiments. Under superinfection conditions, *R. bellii* was allowed to establish infection three days prior to a change in media and gentamicin treatment in association with *A. marginale* infection, while in both coinfection conditions the bacterial stocks were thawed and applied directly to the host cells. More bacterial survival would be expected from an established infection. This may also explain the different *R. bellii* growth phenotypes in the MOI of ten coinfection and the superinfection. Though both experiments showed a decrease in *R. bellii* between the 24 h and 48 h time points, *R. bellii* in the superinfection recovered to more than twice the 48 h amount by 96 hrs. In the high-MOI coinfection, however, the same recovery did not occur over that period. Even so, the MOI of ten coinfection maintained a living population of *R. bellii* equivalent to, or larger than, the 96 h time point of the MOI of one coinfection.

## 5. Conclusions

Bovine anaplasmosis is a major disease of cattle around the world. Despite its high prevalence and economic impact, there are few effective strategies for the treatment and prevention of the disease. Our previous work indicates a negative correlation between the presence and amount of *R. bellii* and the ability of the *D. andersoni* tick to acquire *A. marginale*. This work supports that correlation and indicates a causative relationship. When *R. bellii* was allowed to establish infection in DAE100 cells before *A. marginale* was introduced, *A. marginale* was unable to replicate in tick cells. Moreover, the presence of *R. bellii* reduced the amount of *A. marginale* in the host cells at 24 h. These findings suggest that the presence of *R. bellii* results in decreased entry of *A. marginale* and that an established infection, or very high infection levels of *R. bellii*, affect *A. marginale* growth. Whether these effects are due to a direct interaction between the two bacteria or to the compound effects of host cell manipulation by the two bacterial species is yet to be elucidated. The interaction between these two bacteria highlights the potential effects of arthropod endosymbionts on pathogen transmission by ticks. This may in turn lead to the development of a biocontrol agent for *A. marginale* transmission or lead to the discovery of a novel mechanism for prevention of host cell entry by *A. marginale* that can be exploited for disease prevention.

## Figures and Tables

**Figure 1 microorganisms-11-01096-f001:**
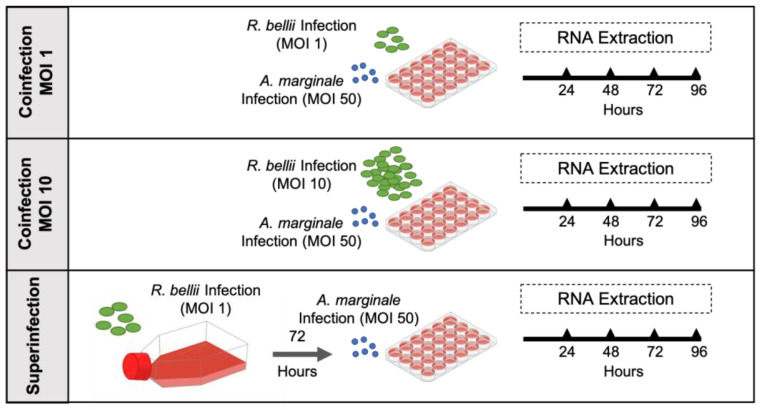
Overview of the three mixed-infection experiments: This figure illustrates the three growth-curve experiments, including different amounts of *R. bellii* and time points of infection for RNA collection. *Anaplasma marginale* was used at an MOI of 50 in all experiments.

**Figure 2 microorganisms-11-01096-f002:**
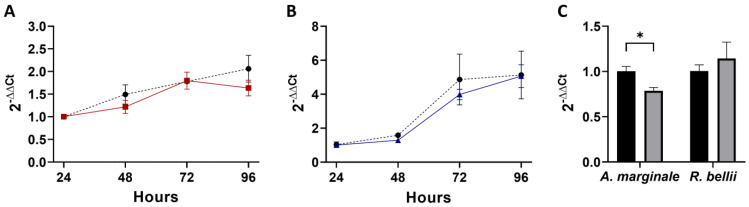
Growth and acquisition data for the low-*R. bellii* coinfection: (**A**) corresponds to the growth curve for *A. marginale* in both the *A. marginale* single-infection control (red) and the coinfection (black). (**B**) is the growth curve of *R. bellii* in the *R. bellii* single-infection control (blue) and the coinfection (black). The ΔΔCt values for both (**A**,**B**) are normalized to the host cell RNA and to the 24 h time point from the same growth condition and the significance of differences between conditions was analyzed with two-way ANOVAs. (**C**) is the amount of *A. marginale* and *R. bellii* present at the 24 h time point of the coinfection (gray), normalized to the host cell RNA, and to the 24 h time point of the corresponding single-infection control (black). Twenty-four-hour time points were compared using two-tailed *t*-tests. Error bars indicate standard error of the mean. * Represents a *p* value below 0.05.

**Figure 3 microorganisms-11-01096-f003:**
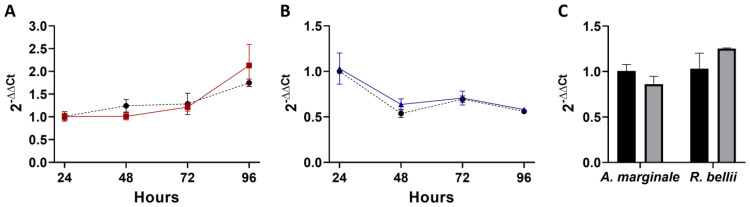
Growth and acquisition data for the high-*R. bellii* coinfection: (**A**) corresponds to the growth curve for *A. marginale* in both the *A. marginale* single-infection control (red) and the coinfection (black). (**B**) is the growth curve of *R. bellii* in the *R. bellii* single-infection control (blue) and the coinfection (black). The ΔΔCt values for both (**A**,**B**) are normalized to the host cell RNA and to the 24 h time point from the same growth condition and the significance of the differences between conditions was analyzed with two-way ANOVAs. (**C**) is the amount of *A. marginale* and *R. bellii* present at the 24 h time point of the coinfection (gray), normalized to host cell RNA, and to the 24 h time point of the corresponding single-infection control (black). Twenty-four-hour time points were compared with two-tailed *t*-tests. Error bars indicate standard error of the mean.

**Figure 4 microorganisms-11-01096-f004:**
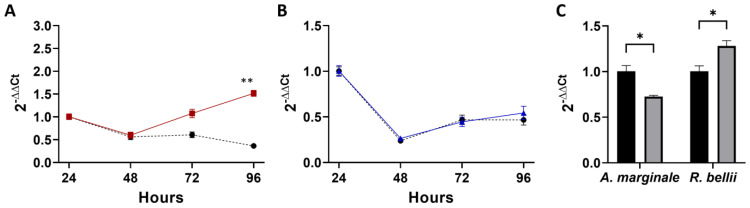
Growth and acquisition data for the superinfection: (**A**) corresponds to the growth curve for *A. marginale* in both the *A. marginale* single-infection control (red) and the coinfection (black). (**B**) is the growth curve of *R. bellii* in the *R. bellii* single-infection control (blue) and the coinfection (black). The ΔΔCt values for both (**A**,**B**) are normalized to the host cell RNA and to the 24 h time point from the same growth condition and the significance of the differences between conditions was analyzed with two-way ANOVAs. Šídák’s multiple-comparison test was used in instances where ANOVA showed significance. (**C**) is the amount of *A. marginale* and *R. bellii* present at the 24 h time point of the coinfection (gray), normalized to the host cell RNA, and to the 24 h time point of the corresponding single-infection control (black). Twenty-four-hour time points were compared with two-tailed *t*-tests. Error bars indicate standard error of the mean. * Represents a *p* value below 0.05 ** Represents a *p* value below 0.01.

**Figure 5 microorganisms-11-01096-f005:**
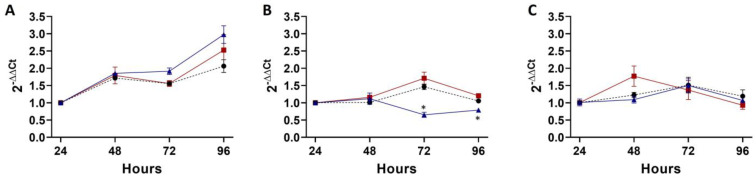
DAE100 cell growth curves across three experimental conditions: (**A**) is the DAE100 growth curve for the low-*R. bellii* coinfection across all experimental conditions, (**B**) is the growth curve of DAE100 cells in the high-*R. bellii* coinfection, and (**C**) is the DAE100 cell growth curve for the superinfection. The ΔΔCt values for all growth curves are normalized to the genomic DNA positive control to facilitate comparison across samples and to the 24 h time point from the same growth condition. All growth curves include the *R. bellii* single-infection control (blue), the *A. marginale* single-infection control (red), and the combined infection (black). The significance of the differences between conditions was analyzed with two-way ANOVAs. Tukey’s test was used in instances where ANOVAs showed significance. Error bars indicate standard error of the mean. * represents a *p* value below 0.05.

**Figure 6 microorganisms-11-01096-f006:**
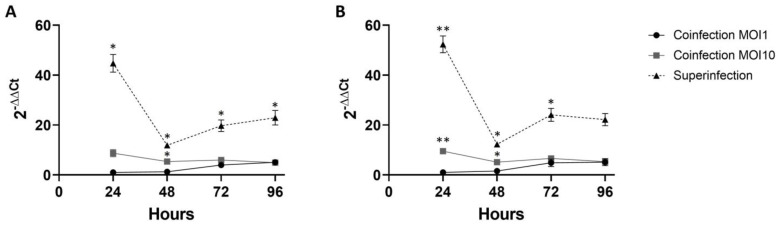
*R. bellii* growth in three experiments normalized to the MOI of 1 infection at 24 h time point: (**A**) is the growth curves for the *R. bellii* single-infection controls from the three experiments. (**B**) is the mixed infection from each of the three experiments. Both panels plot the growth of *R. bellii* normalized to host cell RNAand then to the average of the 24 h time points for the coinfection at an MOI of one. The significance of the differences between conditions was analyzed with two-way ANOVAs. Tukey’s test was used in instances where ANOVA showed significance. Error bars indicate standard error of the mean. * represents a *p* value below 0.05 ** represents a *p* value below 0.01.

**Table 1 microorganisms-11-01096-t001:** Primer sets for RT-qPCR.

Species	Gene	Forward	Reverse	Amplicon Size (bp)
*D. andersoni*	GAPDH	ggtcatctctgctccatctg	tgctcacaatcttcatgcttg	89
*A. marginale*	*msp5*	cttccgaagttgtaagtgagggca	cttatcggcatggtcgcctagttt	203
*R. bellii*	*rpoB*	gcttaaagatcgcaaagggattatagacg	cctgccgacattctttcaactactg	144

## Data Availability

The data presented in this study are available in the body of this article, as well as Appendix A.

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
