# Peer review of "The Effect of Rickettsia bellii on Anaplasma marginale Infection in Dermacentor andersoni Cell Culture"

_microorganisms, 2023, doi:10.3390/microorganisms11051096_

Round 1
Reviewer 1 Report
The manuscript by Aspinwall et al., present data on The Effect of Rickettsia bellii on Anaplasma marginale Infection in Dermacentor andersoni Cell Culture. Overall, the review is well described, and the information is interesting and important for researchers in the field of Anaplasma and ticks.
Only minor revisions are suggested to improve the manuscript.
Line 85 (primer table) include the table number.
Please include the PCR product size for qPCR Table. 1
Author Response
Reviewer 1 asked for only minor revisions:
1) Line 85 (primer table) include the table number.
This has been corrected.
2) Please include the PCR product size for qPCR Table. 1
We have added the size of the amplicons to Table 1.
Reviewer 2 Report
The manuscript is well written and clearly organised, but my main concern is the lack of consistent growth observed for R. belli. To me the results are inconclusive and the conclusions inconsequent until that issue is solved.
Line 104: Why this system used? it´s not clear in the methods
Line 118, 120: Effect of antibiotics on Anaplasma infection?
Figure 2C: Although not significant, R. belli appears to grow better, not worse, in the presence of Anaplasma, why is that? how about at 48, 72 and 96h? what happens then?
Figure 3B: R. belli does not appear to be growing, in fact it looks like its dying, both in the presence and absence of Anaplasma coinfection, this experiment should be repeated.
Figure 4B: Again, R. belli is not growing properly, the results cannot be trusted since the growth of R. belli is unreliable. What happens at 48, 72, 96 h?
Line 234: this is confusing, this section is about DAE100 growth curves or or bacterial growth?
Figure 5. Are the primers the same for this results, using either RNA or genomic DNA? This is not clear. It would be useful to add legends to the graphs, for example blue: R. belli, red: A. marginale, black: combined.
Line 262: it doesn’t look like R. belli infection was well established
Line 273: how about the presence of antibiotics?
Line 302: but R. belli does not replicate properly at high MOI
Author Response
We thank the reviewer for their comments and have responded to each below.
1) The manuscript is well written and clearly organised, but my main concern is the lack of consistent growth observed for R. belli. To me the results are inconclusive and the conclusions inconsequent until that issue is solved.
The reviewer has focused on growth of R. belii, but this study is really focused on the growth of A. marginale, and R. bellii can be viewed as a treatment. We are looking at how the presence of R. bellii affects the ability of A. marginale to enter and grow in the tick cells.
2) Line 104: Why this system used? it´s not clear in the methods
The statement has been modified as follows:
“Plates were incubated at 34ËšC for 1 hr in a sealed EZ Campy container system (Fisher Scientific) to maintain a microaerophilic environment appropriate for A. marginale growth.”
3) Line 118, 120: Effect of antibiotics on Anaplasma infection?
Antibiotics are used to kill the extracellular bacteria. This has been added to the manuscript.
4) Figure 2C: Although not significant, R. belli appears to grow better, not worse, in the presence of Anaplasma, why is that? how about at 48, 72 and 96h? what happens then?
Figure 2C compares the amount of R. bellii that entered the host cell in the presence of A. marginale as compared to when no A. marginale is present. The difference between these is not statistically significant. This is not growth. It is possible that Anaplasma is facilitating more effective entry by R. bellii into host cells, but, again, this is not significantly different.
The growth of R. bellii over time is shown in 2B, and is not significantly different with or without Anaplasma.
5) Figure 3B: R. belli does not appear to be growing, in fact it looks like its dying, both in the presence and absence of Anaplasma coinfection, this experiment should be repeated.
As explained above, R. bellii should be viewed as a treatment variable - we are examining A. marginale growth. The reason R. bellii declines is because a very high number of R. belli were used in the experiment, - at 10 MOI, not all can survive, however, this does not mean that the bacteria that are able to establish in the host are dying. The growth curve is actually more or less flat after 48 hours, indicating that once the number of R. bellii have equilibrated to what the culture can maintain, the number of bacteria were maintained.
6) Figure 4B: Again, R. belli is not growing properly, the results cannot be trusted since the growth of R. belli is unreliable. What happens at 48, 72, 96 h?
In this experiment, there is a drop in the initial phase, but after the 48 hr time point, the R. bellii cells begin to grow. This experiment is conducted with an intermediate amount of R. bellii compared to exp 1 and 2. These R. bellii growth results, with respect to the previous 2 experiments are also intermediate.
It is not clear to me why the reviewer is asking what happens at 48, 72 and 96 hr, as this data is provided in the figure.
7) Line 234: this is confusing, this section is about DAE100 growth curves or or bacterial growth?
This section exclusively looks at DAE100 (host cell growth). The use of bacterial names is simply to clarify what experiment the DAE100 cell data comes from. This was not a distinct set of experiments, but an analysis of the DAE100 cell data from the experiments above.
8) Figure 5. Are the primers the same for this results, using either RNA or genomic DNA? This is not clear. It would be useful to add legends to the graphs, for example blue: R. belli, red: A. marginale, black: combined.
These results are based on RNA extraction and were used to normalize bacterial RNA in the three sections above. The primers are included in the primer table above as well. To help clarify this misunderstanding, section 3.4 was modified slightly.
The requested information is present in the figure legend.
9) Line 262: it doesn’t look like R. belli infection was well established
To clarify, a higher amount of R. bellii is present in the superinfection, and MOI 10 coinfection. The lack of growth is associated with too much, not too little, R. bellii. The "established" infection is a reference to the fact that R. bellii has had an opportunity to infect host cell prior the introduction of A. marginale.
10) Line 273: how about the presence of antibiotics?
Antibiotics were used consistently between experiments and controls. Additionally, after the introduction of A. marginale, gentamicin was used for 1 hour to kill extracellular bacteria (consistent with multiple studies cited in this work), the antibiotic was removed and no antibiotic was used for the rest of the time. An addition to the methods has been made to clarify.
11) Line 302: but R. belli does not replicate properly at high MOI
This study is not about the replication of R. belli, but rather the presence of R. belli in the culture. When supersaturating amounts of R. bellii are used, there is some cell death, but that does not mean that the cells that remain are dead. Particularly because the cell numbers are relative to the host cell number. If the bacteria grow at the same rate as the host cell, they will appear to not be growing in this assay.
Reviewer 3 Report
This work aims to understand factors that influence the correlation between R. bellii and A. marginale in the microbiome of a population of D. andersoni ticks. The authors concluded that A. marginale is less able to establish infection in the presence of R. bellii, and that established R. bellii infection inhibits A. marginale replication.
This research is well written, but the text lacks information of why this research is important and what future applications it might have. This manuscript will benefit for some changes in the abstract, introduction and conclusion.
Abstract: Lacks a strong conclusion. What are the novelty and implication of this study?
Keywords: add Tick-borne diseases
Introduction:
For a higher impact: The introduction will benefit from a sentence and a couple of recent references that address the epidemiology of bovine anaplasmosis around the world, showing it impact (corroborating your conclusion).
The authors indicate that bovine anaplamosis is a major disease of cattle around the world but don’t make any reference to its associated cost and prevalence around the world.
Also, the authors state that there are few effective strategies for treatment and prevention of bovine anaplamosis, but then they do not indicate the relevance of this work. For example, will this work have similar impact as the work with wolbachia that authors referred? Will this interaction between R. bellii and A. marginale help to control of bovine anaplamosis?
Materials and Methods:
Line 68: references 15 and 16 can be together at the end of the sentence.
All text: authors need to make sure that references are all at the end of the sentence; currently is inconsistent.
Line 98: three min – change for 3 min to be consistent throughout the text. Check all text and confirm writing of time. Again, check line 117.
Line 146: times points or time points or timepoints (such in Figure1)?
Lines 149-154 and results: The authors referred statistical analysis with ANOVA and two-tailed t-test, however the results section doesn’t specify which results are related to each test. Also, authors only provide approximate p values when there was significance, but not when there were not significant differences. Could you please provide complete information for all the cases (table)? Perhaps, results of all analysis could be added as a supplement. Were the assumptions for ANOVA and t test verified?
Results
Lines 157 to 167 seems to be part of the materials and methods and not results. The same for Figure 1. Change to the previous section or explain why inclusion as results.
Line 182 (Figure 2): *Represents a - change this in all Figures.
Figures 2, 3, 4 and 5: increase size Y-axis label.
Lines 197-198 and then all text: be consistent when writing numbers (e.g., 3 or three) when in the middle of a sentence.
All figures and correspondent text: if figure is A, the legend will have (A) and not (a); the same for Figure 2a – change for Figure 2A. Change accordingly throughout all text including Figure legends.
Conclusions: Finish with scientific contribution of this work; why this discovery is important.
Data Availability Statement: The authors didn’t provide their own statement. Is the data available somewhere? Can we replicate these results?
Author Response
We thank the reviewer for their very specific comments for greater clarity. We have responded to each below.
1) Abstract: Lacks a strong conclusion. What are the novelty and implication of this study?
We have added a concluding statement to the abstract:
“This interaction highlights the importance of the microbiome in preventing tick vector competence and may lead to the development of a biological or mechanistic control for A. marginale transmission by the tick.”
2) Keywords: add Tick-borne diseases
Done
3) For a higher impact: The introduction will benefit from a sentence and a couple of recent references that address the epidemiology of bovine anaplasmosis around the world, showing it impact (corroborating your conclusion).
The authors indicate that bovine anaplamosis is a major disease of cattle around the world but don’t make any reference to its associated cost and prevalence around the world.
We have added to the introduction and included additional references that address significance and epidemiology.
4) Also, the authors state that there are few effective strategies for treatment and prevention of bovine anaplamosis, but then they do not indicate the relevance of this work. For example, will this work have similar impact as the work with wolbachia that authors referred? Will this interaction between R. bellii and A. marginale help to control of bovine anaplamosis?
This has been addressed by adding a statement to the abstract (see above) and the conclusion at the ending of the paper.
5) Line 68: references 15 and 16 can be together at the end of the sentence.
Done
6) All text: authors need to make sure that references are all at the end of the sentence; currently is inconsistent.
Have never seen this as a requirement, but have made these changes as requested.
7) Line 98: three min – change for 3 min to be consistent throughout the text. Check all text and confirm writing of time. Again, check line 117.
Done
8) Line 146: times points or time points or timepoints (such in Figure1)?
Corrected
9) Lines 149-154 and results: The authors referred statistical analysis with ANOVA and two-tailed t-test, however the results section doesn’t specify which results are related to each test. Also, authors only provide approximate p values when there was significance, but not when there were not significant differences. Could you please provide complete information for all the cases (table)? Perhaps, results of all analysis could be added as a supplement. Were the assumptions for ANOVA and t test verified?
We appreciate this feedback. Therefore, we have added a supplementary file showing the statistical outputs for all analysis. As for verification of the normal distributions, normality is being assumed in respect to variation between the growth of bacteria in replicated conditions. This is commonly accepted to be a normal function.
10) Lines 157 to 167 seems to be part of the materials and methods and not results. The same for Figure 1. Change to the previous section or explain why inclusion as results.
Though this section contains background information consistent with the materials and methods, we felt it was important to reorient readers to the experimental design early in the results, particularly when the set of experiments are so closely related.
11) Line 182 (Figure 2): *Represents a - change this in all Figures.
Done
12) Figures 2, 3, 4 and 5: increase size Y-axis label.
Done
13) Lines 197-198 and then all text: be consistent when writing numbers (e.g., 3 or three) when in the middle of a sentence.
We have followed the convention of numbers less than 10 should be spelled out, unless being followed by a unit of measure. We have checked for consistency.
14) All figures and correspondent text: if figure is A, the legend will have (A) and not (a); the same for Figure 2a – change for Figure 2A. Change accordingly throughout all text including Figure legends.
Corrected.
15) Conclusions: Finish with scientific contribution of this work; why this discovery is important.
Done.
16) Data Availability Statement: The authors didn’t provide their own statement. Is the data available somewhere? Can we replicate these results?
Done.
Reviewer 4 Report
Following up on earlier microbiome work showing that presence of R. bellii in Rocky Mountain wood ticks reduced their ability to acquire Anaplasma marginale infection, in this paper the authors study this potential competition effect in a tick cell culture experimental system. This overcomes the confounding effects of other microbiome interactions that would be encountered in vivo, allowing the interactions of the two organisms of interest to be studied in greater detail.
Using well-controlled coinfection and superinfection experiments, this study provides supporting evidence that R. bellii presence is linked to reduced ability of A. marginale to establish infection in ticks. The potential mechanisms by which this may be achieved are discussed and it should be interesting to look into these with future research.
While based on a relatively small set of experiments, this study adds further evidence on the role of symbiont-pathogen relationships in tick vector competence, and therefore makes an important contribution to the field.
Thus I recommend that the paper be accepted with minor corrections:
line 85: "primer table" should reference Table 1.
line 91: change "than" to "then"
line 105: clarify why gentamicin was used - to kill any remaining extracellular bacteria?
line 138/Table 1: check - should the gene name be GAPDH rather than GltA?
line 256: replace "or" with "of"
A recent study also showed competition between symbiotic Rickettsia and Anaplasma phagocytophilum in tick cell culture, and this work could be mentioned in the Discussion and/or Introduction as it appears to be the only other study looking at microbial competition in tick cells, as well as being highly relevant to this manuscript:
Front Vet Sci. 2022 Jan 6;8:748427.
doi: 10.3389/fvets.2021.748427.
The Ixodes scapularis Symbiont Rickettsia buchneri Inhibits Growth of Pathogenic Rickettsiaceae in Tick Cells: Implications for Vector Competence
Author Response
We thank the reviewer for the knowledgeable review.
We have incorporated all of the suggested changes/additions.
Round 2
Reviewer 2 Report
- Can you provide more evidence of a well established infection/presence, like a fluorescence/confocal image since R belli expresses GFP?
- "This study is not about the replication of R. belli, but rather the presence of R. belli in the culture. When supersaturating amounts of R. bellii are used, there is some cell death, but that does not mean that the cells that remain are dead. Particularly because the cell numbers are relative to the host cell number. If the bacteria grow at the same rate as the host cell, they will appear to not be growing in this assay."
I don´t agree with this statement, I did not focus on R. belli, I do realize this paper is about Anaplasma growth but if the basis of this work was to compare Anaplasma growth alone vs co-infections with R. belli, I fail to see to value of the experiments if they cannot guarantee a well established presence of R. belli. On top of that, tick cells replicate very slowly, the opposite is more likely to occur.
Reviewer 3 Report
Thank you for including all the suggestions and answering the questions from all reviewers.
Author Response
Thank you.